# A comment on instantons and their fermion zero modes in adjoint $QCD_2$.

## A.V. Smilga

*SUBATECH, Université de Nantes, 4 rue Alfred Kastler, BP 20722, Nantes 44307, France.*

### Abstract

The adjoint 2-dimensional $QCD$ with the gauge group $SU(N)/Z_N$ admits topologically nontrivial gauge field configurations associated with nontrivial $\pi_1[SU(N)/Z_N] = Z_N$. The topological sectors are labelled by an integer $k = 0, \ldots, N - 1$. However, in contrast to $QED_2$ and $QCD_4$, this topology is not associated with an integral invariant like the magnetic flux or Pontryagin index. These instantons may admit fermion zero modes, but there is always an equal number of left-handed and right-handed modes, so that the Atiyah-Singer theorem, which determines in other cases the number of the modes, does not apply.

The *mod. 2 argument* [1] suggests that, for a generic gauge field configuration, there is either a single doublet of such zero modes or no modes whatsoever. However, the known solution of the Dirac problem for a wide class of gauge field configurations [2, 3, 4] indicates the presence of $k(N - k)$ zero mode doublets in the topological sector $k$. In this note, we demonstrate in an explicit way that these modes are not robust under a generic enough deformation of the gauge background and confirm thereby the conjecture of Ref. [1].

The implications for the physics of this theory (screening vs. confinement issue) are briefly discussed.

# 1 Introduction

The Lagrangian of the massless 2-dimensional QCD with fermions lying in the adjoint representation of the $SU(N)$ gauge group reads [5]

$$\mathcal{L} = \text{Tr}\left\{-\frac{1}{2}F_{\mu\nu}F_{\mu\nu} + i\bar{\psi}\gamma_\mu D_\mu\psi\right\}, \tag{1.1}$$

where $A_\mu = A_\mu^a t^a$, $\psi = \psi^a t^a$ is a 2-component Majorana spinor, $D_\mu\psi = \partial_\mu\psi - ig[A_\mu, \psi]$, and the coupling $g$ has the dimension of mass. We will not display the dependence on $g$ in what follows. It can always be restored on dimensional grounds. In 2-dimensional Minkowski space, the gamma matrices can be chosen as $\gamma_M^0 = \sigma^2, \gamma^1 = i\sigma_1$, where $\sigma_{1,2,3}$ are the standard Pauli matrices. Then $\gamma^5 = \gamma_M^0\gamma_M^1 = \sigma_3$. The Euclidean gamma matrices, which we will mostly need in the following, are

$$\gamma_E^0 = \sigma_2, \qquad \gamma^1 = \sigma_1. \tag{1.2}$$

In this case, the elements of the center of $SU(N)$ do not act faithfully on the fields, and we are dealing with the group $SU(N)/Z_N$. Its fundamental group is nontrivial, $\pi_1[SU(N)/Z_N] = Z_N$, which leads to the existence of topologically nontrivial Euclidean field configurations—the instantons [6, 2]. They are characterized by an integer $k = 0, 1, \ldots, N-1$.

In Refs. [2, 3], we imposed certain natural boundary conditions for the fermion fields and solved the Dirac equation for a class of topologically nontrivial backgrounds on a cylinder [2] and on the Euclidean plane [3]. We found the existence of $k(N-k)$ left-handed and $k(N-k)$ right-handed zero modes in the spectrum. It was, however, argued in Ref.[1] that, for a generic field configuration, there are only $[k(N-k)]_{\text{mod.2}}$ doublets of zero modes.

The main motivation of this study was to reconcile these two seemingly contradictive statements. As a result, we confirm the conjecture of Ref. [1] and demonstrate how the *naive* zero modes disappear when the background is deformed in a general enough way.

The plan of the paper is the following.

In the next section, we consider, following Ref. [1], the theory on a finite torus and show that, even in a topologically nontrivial sector, the field density can be brought to zero by a continuous deformation of the potential.

In Sect. 3, we study the spectrum of the Dirac operator and show that all its eigenstates with nonzero eigenvalues $\lambda$ are split the quartets including two degenerate states with the eigenvalue $|\lambda|$ and two degenerate states with the eigenvalue $-|\lambda|$. The spectrum may include also some number of the zero mode doublets. This number may be odd, and in this case at least one such doublet must stay on zero for an arbitrary deformation, or it can be even, and in this case the zero modes are not protected. This generic *mod. 2 argument* is confirmed by the analysis of the topologically nontrivial toroidal configuration with zero field density. The number of the zero modes depends on the fermionic boundary conditions. Choosing the conditions similar to the conditions in Refs. [2, 3, 4], one finds in this case that the zero modes are absent if $N$ is odd and there are $\gcd(N, k)$ doublets of zero modes if $N$ is even. This does not coincide with $k(N-k)$, but one can observe that the two estimates have the same *parity*.

In Sect. 4, we consider, following [2, 3, 4], the theory on a cylinder $S^1 \times R$, with $S^1$ being a finite spatial circle with antiperiodic boundary conditions for the fermion fields and $R$ the infinite Euclidean time axis, $t \in (-\infty, \infty)$. For a special *Cartan* instanton background,

the Dirac operator admits in this case $k(N-k)$ doublets of zero modes. The same result is reproduced for the theory placed on a finite torus.

In Sect. 5, we first show that these zero modes are robust under a particular class of deformations that do not modify the values of the potential at $t = \pm\infty$. Then we consider more general deformations of the potential and show that in this case most zero modes disappear, leaving only one doublet of the modes when $k(N-k)$ is odd.

In Sect. 6, we discuss the impact of the zero analysis presented in the main body of the paper on the physics of this theory — whether it exhibits *confinement* with the area law for the fundamental Wilson loops or *screening* characterized by the perimeter law.

# 2 Instantons on the torus

We consider the theory on an Euclidean torus,

$$0 \le x \le L, \qquad 0 \le t \le \beta. \tag{2.1}$$

The generic toroidal boundary conditions for $A_\mu(x,t)$ read [7]

$$
\begin{aligned}
A_\mu(x+L,t) &= A_\mu^{\Omega_1}(x,t) \equiv -i[\partial_\mu\Omega_1(x,t)]\Omega_1^{-1}(x,t) + \Omega_1(x,t)A_\mu(x,t)\Omega_1^{-1}(x,t), \\
A_\mu(x,t+\beta) &= A_\mu^{\Omega_2}(x,t) \equiv -i[\partial_\mu\Omega_2(x,t)]\Omega_2^{-1}(x,t) + \Omega_2(x,t)A_\mu(x,t)\Omega_2^{-1}(x,t). 
\end{aligned}
\tag{2.2}
$$

However, the gauge transformation matrices $\Omega_{1,2} \in SU(N)$ are not quite arbitrary: we have to require that, going from the point $(x,t)$ to the point $(x+L, t+\beta)$ in two different ways, we obtain the same potential $A_\mu(x+L, t+\beta)$. Bearing in mind that the potential does not transform under the action of the center, we obtain the self-consistency condition

$$\Omega_1(x,t+\beta)\,\Omega_2(x,t) = \omega_N^k\,\Omega_2(x+L,t)\,\Omega_1(x,t) \tag{2.3}$$

with

$$\omega_N = e^{2\pi i/N}.$$

An integer $k$ labels the topological sector.

We choose in the following $\Omega_1(x,t) = \mathbb{1}$ and the time-independent $\Omega_2(x,t) \equiv \Omega(x)$ that interpolates between $\Omega(0) = \mathbb{1}$ and $\Omega(L) = \omega_N^k\mathbb{1}$. This gives

$$
\begin{aligned}
A_\mu(x+L,t) &= A_\mu(x,t), \\
A_\mu(x,t+\beta) &= -i[\partial_\mu\Omega(x)]\Omega^{-1}(x) + \Omega(x)A_\mu(x,t)\Omega^{-1}(x), 
\end{aligned}
\tag{2.4}
$$

Note that $\Omega(x)$ represents a noncontractible loop in $SU(N)/Z_N$. It is a "large" gauge transformation.

**Theorem 1.** *[8] The field density of any field configuration $A_\mu(x,t)$ satisfying (2.4) can be brought to zero by a smooth deformation.*

*Proof.* Consider first the simplest case **N = 2**. By a topologically trivial gauge transformation, we can bring $\Omega(x)$ to the form[1]

$$\Omega(x) = \exp\left\{\frac{i\pi x}{L}\tau^3\right\}. \tag{2.5}$$

---

[1] $\tau^a = 2t^a$ are the Pauli matrices in the color space.

Perform now a gauge transformation

$$A_\mu \;\to\; B_\mu = -i(\partial_\mu U)U^{-1} + U A_\mu U^{-1} \tag{2.6}$$

with $U(x,t)$ satisfying the following b.c. :

$$
\begin{aligned}
U(x+L,t) &= i\tau^1 U(x,t), \\
U(x,t+\beta) &= i\tau^3 U(x,t)\Omega^{-1}(x)\,.
\end{aligned} \tag{2.7}
$$

**Lemma.** *A continuous matrix function satisfying (2.7) exists.*

*Proof.* We choose $U(0,0) = \mathbb{1}$. Then the conditions (2.7) dictate the following values of $U(x,t)$ at the edges of the square:

$$
\begin{aligned}
U(x,0) &= \exp\left\{\frac{i\pi x}{2L}\tau^1\right\}, \\
U(0,t) &= \exp\left\{\frac{i\pi t}{2\beta}\tau^3\right\}, \\
U(x,\beta) &= i\tau^3 \exp\left\{\frac{i\pi x}{2L}\tau^1\right\}\exp\left\{-\frac{i\pi x}{L}\tau^3\right\}, \\
U(L,t) &= i\tau^1 \exp\left\{\frac{i\pi t}{2\beta}\tau^3\right\}.
\end{aligned} \tag{2.8}
$$

In the corners:

$$U(0,0) = \mathbb{1}, \quad U(0,\beta) = i\tau^3, \quad U(L,0) = i\tau^1, \quad U(L,\beta) = i\tau^2\,.$$

Capitalizing on the fact that $\pi_1[SU(2)] = 0$, we can also continuously define $U(x,t)$ in the interior of the square. $\qquad\square$

If $A_\mu(x,t)$ satisfies (2.4) and $U(x,t)$ satisfies (2.7), then the gauge-transformed field $B_\mu(x,t)$ satisfies

$$
\begin{aligned}
B_\mu(x+L,t) &= \tau^1 B_\mu(x,t)\tau^1\,, \\
B_\mu(x,t+\beta) &= \tau^3 B_\mu(x,t)\tau^3\,.
\end{aligned} \tag{2.9}
$$

These b.c. allow for a continuous deformation $B_\mu \to 0$. Performing the inverse gauge transformation with the matrix $U^{-1}(x,t)$, we obtain a pure gauge configuration,

$$A_\mu^{\text{deformed}}(x,t) = iU^{-1}\partial_\mu U\,. \tag{2.10}$$

The corresponding field density is zero.

Let now **N = 3**. There are two nontrivial topological sectors: with $k = 1$ and with $k = 2$. If $k = 1$, the loop $\Omega(x)$, uncontractible in $SU(3)/Z_3$, can be chosen in the form

$$\Omega(x) = \exp\left\{\frac{2i\pi x}{3L}\text{diag}(1,1,-2)\right\}, \tag{2.11}$$

so that $\Omega(L) = \omega_3\mathbb{1}$.

Perform the gauge transformation (2.6) with $U(x,t)$ satisfying the following b.c. :

$$
\begin{aligned}
U(x+L,t) &= SU(x,t)\,, \\
U(x,t+\beta) &= CU(x,t)\Omega^{-1}(x)\,,
\end{aligned}
\tag{2.12}
$$

where $C, S \in SU(3)$ are the "clock and shift" matrices [9]:

$$
C = \begin{pmatrix} 1 & 0 & 0 \\ 0 & \omega_3 & 0 \\ 0 & 0 & \omega_3^2 \end{pmatrix}, \qquad S = \begin{pmatrix} 0 & 0 & 1 \\ 1 & 0 & 0 \\ 0 & 1 & 0 \end{pmatrix}.
\tag{2.13}
$$

The identities

$$
S^3 = C^3 = 1, \quad SC = \omega_3^2 CS
$$

hold.

One can represent $C = e^{iP}$, $S = e^{iQ}$ with Hermitian $P, Q$. On the edges of the square, we derive

$$
\begin{aligned}
U(x,0) &= \exp\left\{\frac{ixQ}{L}\right\}, \\
U(0,t) &= \exp\left\{\frac{iPt}{\beta}\right\}, \\
U(x,\beta) &= C\exp\left\{\frac{iQx}{L}\right\}\Omega^{-1}(x)\,, \\
U(L,t) &= S\exp\left\{\frac{iPt}{\beta}\right\}.
\end{aligned}
\tag{2.14}
$$

In the corners:

$$
U(0,0) = \mathbb{1}, \quad U(0,\beta) = C, \quad U(L,0) = S, \quad U(L,\beta) = SC\,.
$$

Bearing in mind that $\pi_1[SU(3)] = 0$, we can also continuously define $U(x,t)$ in the interior of the square.

The gauge-transformed field (2.6) satisfies the conditions

$$
\begin{aligned}
B_\mu(x+L,t) &= SB_\mu(x,t)S^{-1}\,, \\
B_\mu(x,t+\beta) &= CB_\mu(x,t)C^{-1}\,.
\end{aligned}
\tag{2.15}
$$

It can be smoothly deformed to zero, which means that the original field $A_\mu(x,t)$ satisfying the conditions (2.4) can be smoothly deformed to a pure gauge form with zero field density.

In the sector $k = 2$, we choose

$$
\Omega(x) = \exp\left\{\frac{2i\pi x}{3L}\mathrm{diag}(2,-1,-1)\right\},
\tag{2.16}
$$

If we substitute $\omega_3 \to \omega_3^{-1} = \omega_3^2$ in the definition of the matrix $C$ and the subsequent formulas, the whole reasoning can be repeated.

**N ≥ 4**. The generalization is straightforward. In the sector with a given $k$, one should choose $\Omega(x)$ in the form

$$\Omega(x) \;=\; \exp\left\{\frac{2i\pi x}{NL}\mathrm{diag}(\underbrace{k,\ldots,k}_{N-k},\ \underbrace{k-N,\ldots,k-N}_{k})\right\},\tag{2.17}$$

so that $\Omega(L) = \omega_N^k \mathbb{1}$ with $\omega_N = e^{2i\pi/N}$.

The matrices $C, S$ entering the boundary conditions (2.12) and (2.15) may now be defined as

$$
\begin{aligned}
C &\;=\; e^{i\pi k(N+1)/N}\mathrm{diag}\{1,\omega_N^k,\ldots,\omega_N^{k(N-1)}\}\,,\\[4pt]
S &\;=\; e^{i\pi(N+1)/N}
\begin{pmatrix}
0 & 0 & \cdots & \cdots & 1\\
1 & 0 & \cdots & \cdots & 0\\
\cdots & \cdots & \cdots & \cdots & \cdots\\
0 & \cdots & \cdots & 1 & 0
\end{pmatrix}.
\end{aligned}\tag{2.18}
$$

The phase factors in (2.18) are chosen such that $C, S \in SU(N)$: $\det C = \det S = 1$.

As earlier, the field $B_\mu(x, t)$, untwined by the gauge transformation, can be continuously deformed to zero.

$\square$

The theorem just proven clearly displays that a nontrivial topology is *not* associated in our case with any topological *charge* like the magnetic flux in the 2D Abelian gauge theory or the Pontryagin index in the 4D Yang-Mills theory,[2]

$$q_2 \;=\; \frac{1}{2\pi}\int F,\qquad q_4 \;=\; \frac{1}{8\pi^2}\int \mathrm{Tr}\{F \wedge F\}\,.\tag{2.19}$$

Indeed, the presence of the topological invariants (2.19) depends on the fact that their integrands are exact forms: $F^{\mathrm{Ab}} = dA$ and $\mathrm{Tr}\{F \wedge F\} = d\,\mathrm{Tr}\{A \wedge dA + \frac{2i}{3}A \wedge A \wedge A\}$. But in $QCD_2$, the form $F$ is not exact and $\mathrm{Tr}\{F\}$ is simply zero.

# 3 Dirac operator and its spectrum

Bearing in mind the chosen explicit form (1.2) of the Euclidean gamma matrices, the Euclidean Dirac spectral problem reads

$$i\mathcal{D}\psi = \left(\sigma_2\frac{\partial}{\partial t} + \sigma_1\frac{\partial}{\partial x}\right)\psi - i(\sigma_2[A_0,\psi] + \sigma_1[A_1,\psi]) \;=\; i\lambda\psi\,,\tag{3.1}$$

$\psi \equiv \psi^a t^a$. It has the following important symmetries:

- For any eigenfunction $\psi$ with eigenvalue $\lambda$, the function

$$\psi' = \sigma_3\psi\tag{3.2}$$

is an eigenfunction with eigenvalue $-\lambda$.

---

[2]Here $F$ is the field density 2-form $F = F_{\mu\nu}dx^\mu \wedge dx^\nu/2$.

- For any eigenfunction $\psi$ with eigenvalue $\lambda$, the function

$$\psi'' = \sigma_2 \psi^*  \qquad (3.3)$$

is an eigenfunction with the same eigenvalue.

Suppose $\lambda \neq 0$. Then all the states

$$\psi_1, \quad \psi_2 = \sigma_3 \psi_1, \quad \psi_3 = \sigma_2 \psi_1^*, \quad \psi_4 = \sigma_3 \sigma_2 \psi_1^*$$

are linearly independent. Indeed, the states $\psi_1, \psi_3$ cannot coincide with the states $\psi_2, \psi_4$ because the latter have a different eigenvalue of $\mathcal{D}$. On the other hand, the state $\psi_1$ cannot coincide with $\psi_3$. Indeed, the equality $\sigma_2 \psi_1^* = \kappa \psi_1$ can be spelled out in terms of the spinor components of $\psi_1$ as

$$\begin{pmatrix} 0 & -i \\ i & 0 \end{pmatrix} \begin{pmatrix} a^* \\ b^* \end{pmatrix} = \kappa \begin{pmatrix} a \\ b \end{pmatrix} \implies \begin{cases} ia^* = \kappa b \\ -ib^* = \kappa a \end{cases},$$

which is possible only if $a = b = 0$.

In other words, the spectrum of the excited states of the operator $H = \mathcal{D}^2$ in a generic gauge background is split into quartets of degenerate states. $H$ is a second-order differential operator, which may be called a Hamiltonian. The four-fold degeneracy of all the excited levels means that this Hamiltonian enjoys extended $\mathcal{N} = 2$ supersymmetry. Extended supersymmetry implies the existence of two doublets of Hermitially conjugated supercharges. One such doublet, associated with the symmetry (3.2) has a nice local form,

$$Q = \mathcal{D}(1 + \sigma_3), \qquad Q^\dagger = \mathcal{D}(1 - \sigma_3). \qquad (3.4)$$

Another doublet associated with the symmetry (3.3) is nonlocal.

Two of the quartet states, $\psi_1 + \psi_2$ and $\psi_3 + \psi_4$ are right-handed (they are eigenstates of the chirality operator $\gamma^5 = \sigma^3$ with eigenvalue $+1$) and two other states, $\psi_1 - \psi_2$ and $\psi_3 - \psi_4$ are left-handed.

If $\lambda = 0$, $\psi_1$ and $\psi_2$ are not necessarily linearly independent. They are not if $\psi_1$ has a definite chirality. In this case, we have a doublet of states, $\psi_1$ and $\psi_3$. Note that these states have opposite chiralities, so that the Atiyah-Singer index $n_L - n_R$ of the supersymmetric Hamiltonian $H$ is equal to zero.

Suppose that in a particular gauge background there is only one doublet of the fermion zero modes. Then these modes cannot shift from zero under a smooth deformation, because a doublet cannot become a quartet. But if there are two such doublets, they can move from zero simultaneously, forming a quartet. Similarly, if we have any even number $2n$ of doublets — they all can move from zero forming $n$ quartets of excited states. And if there were an odd number $2n + 1$ of doublets, $2n$ of them can be shifted from zero, but one doublet is bound to stay.

This is the mod. 2 index argument of Ref. [1]. It says that there are no compelling reasons to expect the existence of more than one doublet of the fermion zero modes of the Dirac operator (3.1) in a generic gauge background.

To find out whether this theoretical lower bound for the number of the zero modes is saturated for particular backgrounds, one should perform an explicit study of the solutions to the problem (3.1).

One possibility [1] is to perform a gauge transformation (2.6) of the gauge fields and simultaneously of the fermion fields and then smoothly deform $B_\mu(x,t)$ to zero, in which case we are simply dealing with the free Dirac problem,

$$\left(\sigma_2\frac{\partial}{\partial t} + \sigma_1\frac{\partial}{\partial x}\right)\Psi(x,t) = 0 \tag{3.5}$$

Eq. (3.5) is equivalent to a doublet of equations

$$\left(\frac{\partial}{\partial x} \pm i\frac{\partial}{\partial t}\right)\Psi_\pm(x,t) = 0, \tag{3.6}$$

for the upper and lower spinor components. The solution to (3.6) is very simple: $\Psi_+$ must be holomorphic and $\Psi_-$ antiholomorphic in $z = x + it$.

A nontriviality resides, however, in the boundary conditions to be imposed on the fermion field. The result depends on their choice. We choose the conditions

$$\begin{aligned}
\psi(x+L,t) &= -\psi(x,t), \\
\psi(x,t+\beta) &= \Omega(x)\psi(x,t)\Omega^{-1}(x),
\end{aligned} \tag{3.7}$$

These conditions are similar to (2.4) (periodicity in space and the periodicity up to a large gauge transformation under the imaginary time shift), but note the presence of the extra minus in the first line. We inserted it to make contact with the settings of Ref. [4], where the fermion fields were also chosen to be antiperiodic in spatial direction. This choice can be traced back to earlier papers [10, 2] where the adjoint $QCD_2$ at finite temperature was studied — as is well-known, a finite temperature amounts to a finite Euclidean time extension with antiperiodic conditions for the fermion fields. In [4], the whole picture was rotated by $\pi/2$ and the theory was considered on a finite spatial circle, while keeping the antiperiodic fermion boundary conditions.

Consider first the case **N = 2**. After the gauge transformation

$$\psi(x,t) \to \Psi(x,t) = U(x,t)\psi(x,t)U^{-1}(x,t), \tag{3.8}$$

with $U(x,t)$ satisfying (2.7), the transformed field satisfies the conditions

$$\begin{aligned}
\Psi(x+L,t) &= -\tau^1\Psi(x,t)\tau^1, \\
\Psi(x,t+\beta) &= \tau^3\Psi(x,t)\tau^3,
\end{aligned} \tag{3.9}$$

The matrix functions $\Psi_\pm(x,t)$ are double periodic functions on the large torus, $\{0 \leq x \leq 2L, \ 0 \leq t \leq 2\beta\}$. The only (anti)holomorphic nonsingular double periodic function (the absence of the poles follows from the normalizabily requirement) is a *constant*. We only have to check now if the boundary conditions (3.9) admit constant solutions. The answer is positive, the solution is $\Psi \propto \tau^3$. We have proven the theorem:

**Theorem 2.** *In the $N = 2$ theory with the fermion boundary conditions (3.7), the Dirac operator on the background $A_\mu(x,t)$ representing a pure gauge (2.10) [which corresponds to $B_\mu(x,t) = 0$] has one left-handed and one right-handed zero mode.*

Note that, for the fermion boundary conditions with the positive sign in the first line in (3.7) and (3.9), there would be no zero mode solutions whatsoever.

The next in complexity case is **N = 3**. Let $k = 1$. We may repeat our reasoning by imposing the fermion boundary conditions (3.7) with $\Omega(x)$ given by (2.11), performing the gauge transformation with the parameter $U(x,t)$ satisfying (2.12) and choosing the background $B_\mu(x,t) = 0$. The problem boils down to the search of the (anti)holomorphic Hermitian matrix functions $\Psi(x \pm it)$ that satisfy the conditions

$$
\begin{aligned}
\Psi(x+L,t) &= -S\Psi(x,t)S^{-1}, \\
\Psi(x,t+\beta) &= C\Psi(x,t)C^{-1}.
\end{aligned}
\tag{3.10}
$$

However, such functions do not exist. The conditions (3.10) imply the periodicity in the imaginary time direction and antiperiodicity in the spatial direction on the large torus, $\{0 \le x \le 3L, \ 0 \le t \le 3\beta\}$. The only (anti)holomorphic nonsingular matrix that satisfies this condition is $\Psi(x,t) = 0$. Obviously, this reasoning applies to any odd $N$ with any $k$.

We have proven the theorem:

**Theorem 3.** *In the theory with odd $N$ and with the fermion boundary conditions (3.7) in any topological sector $k$, the Dirac operator on the background $A_\mu(x,t)$ representing a pure gauge (2.10) does not admit zero modes.*

Consider now the case of generic even $N \ge 4$.

We have to search constant Hermitian matrices $\Psi$ that commute with $C$ and anticommute with $S$ in (2.18).

• Consider first the case $k = 1$. Then the condition $\Psi C = C\Psi$ brings $\Psi$ in the Cartan subalgebra. There is only one (up to a factor) diagonal real traceless matrix that anticommutes with $S$:

$$
\Psi = \text{diag}(1, -1, \ldots, 1, -1),
\tag{3.11}
$$

giving a single zero mode of the Dirac operator. Clearly, this also applies to the theory with any $k$ that does not have common nontrivial divisors with $N$.

• Let $k = 2$. The matrix $C$ may now be chosen as

$$
C_{k=2} = \text{diag}(1, e^{4i\pi/N}, \ldots, e^{-4i\pi/N}, 1, e^{4i\pi/N}, \ldots, e^{-4i\pi/N})
$$

Each eigenvalue is repeated twice. The centralizer of such $C$ is the subalgebra

$$
\mathbb{c} = \underbrace{su(2) \oplus \cdots \oplus su(2)}_{N/2} \oplus \underbrace{u(1) \oplus \cdots \oplus u(1)}_{N/2-1}.
\tag{3.12}
$$

This centralizer includes the Cartan subalgebra, which gives the zero mode (3.11) as earlier, and also certain nondiagonal matrices depending on $N/2$ complex parameters. In the particular case $N = 6$, these matrices have the form

$$
\Psi = \begin{pmatrix}
0 & 0 & 0 & a_1 & 0 & 0 \\
0 & 0 & 0 & 0 & a_2 & 0 \\
0 & 0 & 0 & 0 & 0 & a_3 \\
a_1^* & 0 & 0 & 0 & 0 & 0 \\
0 & a_2^* & 0 & 0 & 0 & 0 \\
0 & 0 & a_3^* & 0 & 0 & 0
\end{pmatrix}.
\tag{3.13}
$$

The condition $\Psi S + S\Psi = 0$ implies the chain of $N/2$ relations

$$a_1 + a_2 \;=\; \ldots \;=\; a_{N/2-1} + a_{N/2} \;=\; a_{N/2} + a_1^* \;=\; 0\,. \qquad (3.14)$$

The solution is

$$\{a_1, \ldots, a_{N/2}\} \;=\; \lambda\,\{1, -1, \ldots, (-1)^{N/2-1}\} \qquad (3.15)$$

with a real $\lambda$ if $N/2$ is even and an imaginary $\lambda$ if $N/2$ is odd. This gives the second doublet of zero modes.

We obtain the same result (two zero modes) for any $k$ with $\gcd(N, k) = 2$. In this case, the centralizer of $C$ is still the subalgebra (3.12), the nondiagonal elements of the centralizer are still parametrized by $N/2$ complex numbers $a_j$ and we still have one Cartan zero mode doublet and one doublet (3.15).

 • Let now $\gcd(N, k) = 3$. In this case, the clock matrix $C$ includes $N/3$ different eigenvalues that enter thrice. The centralizer is

$$\mathfrak{c} \;=\; \underbrace{su(3) \oplus \cdots \oplus su(3)}_{N/3} \oplus \underbrace{u(1) \oplus \cdots \oplus u(1)}_{N/3-1}\,. \qquad (3.16)$$

Its nondiagonal elements include $N$ complex parameters organized in two different "ladders", as illustrated below for $N = 6, k = 3$.

$$\Psi \;=\; \begin{pmatrix} 0 & 0 & a_1 & 0 & a_5 & 0 \\ 0 & 0 & 0 & a_2 & 0 & a_6 \\ a_1^* & 0 & 0 & 0 & a_3 & 0 \\ 0 & a_2^* & 0 & 0 & 0 & a_4 \\ a_5^* & 0 & a_3^* & 0 & 0 & 0 \\ 0 & a_6^* & 0 & a_4^* & 0 & 0 \end{pmatrix}. \qquad (3.17)$$

In contrast to (3.13), the ladders in (3.17) are "long" — they involve 6 complex parameters each. On the other hand, the elements in the left ladder and in the right ladder are complex conjugated to each other.

The condition $\{\Psi, S\} = 0$ that the matrix (3.17) should satisfy to represent a zero mode gives the long chain of relations

$$a_1 + a_2 \;=\; a_2 + a_3 \;=\; a_3 + a_4 \;=\; a_4 + a_5^* \;=\; a_5^* + a_6^* \;=\; a_6^* + a_1 \;=\; 0 \qquad (3.18)$$

for the left ladder, and the right ladder gives nothing new. The solution is

$$\{a_1, \ldots, a_6\} \;=\; \{a, -a, a, -a, a^*, -a^*\} \qquad (3.19)$$

with a complex a. For a generic $N$ with $\gcd(N, k) = 3$, the relations are

$$a_1 + a_2 \;=\; \ldots \;=\; a_{2N/3-1} + a_{2N/3} \;=\; a_{2N/3} + a_{2N/3+1}^* \;=\; \ldots \;=\; a_{N-1}^* + a_N^* \;=\; 0\,, \quad (3.20)$$

and their solution also involves a single complex parameter. This gives 2 nondiagonal zero mode doublets, to which the Cartan doublet should be added.

• When $\gcd(N, k) = 4$, the matrix $C$ includes 4 coinciding sets of $N/4$ different eigenvalues, the centralizer is

$$\mathbb{c} = \underbrace{su(4) \oplus \cdots \oplus su(4)}_{N/4} \oplus \underbrace{u(1) \oplus \cdots \oplus u(1)}_{N/4-1}, \tag{3.21}$$

and its nondiagonal elements depend on $3N/2$ complex parameters organized in three ladders: two of them are complex conjugate to each other, depend on $N$ different complex parameters and the condition $\{\Psi, S\} = 0$ gives a "long" chain of relations like in (3.20), leaving only one complex parameter. Besides, there is a ladder depending on $N/2$ complex parameters and their complex conjugates. The requirement $\{\Psi, S\} = 0$ gives a short chain of relations like in (3.14), leaving only one real parameter. We obtain four doublets of zero modes: a Cartan doublet and three nondiagonal doublets.

• This counting is easily generalized for an arbitrary $k$. When $\gcd(N, k) = r$ and $r$ is odd, the nondiagonal elements of the centralizer of $C$ are parameterized by $N(r-1)/2$ complex numbers organized in $r-1$ "long" ladders. Only a half of these ladders are relevant — the other half includes complex conjugated parameters. After imposing the condition $\Psi S + S \Psi = 0$, only one complex parameter is left for each doublet of complex conjugate ladders. This gives $r-1$ doublets of zero modes, to which the Cartan doublet should be added.

If $r$ is even, the nondiagonal part of the centralizer still depends on $N(r-1)/2$ complex parameters organized in $r-1$ ladders. But only $r-2$ of these ladders [$(r-2)/2$ doublets of complex conjugated ladders) are long. They originally include $N(r-2)/2$ complex parameters, of which only $(r-2)/2$ are left after imposing the condition $\{\Psi, S\} = 0$. This gives $r-2$ doublets of zero modes. There is also a "short" ladder depending on $N/2$ complex parameters, of which only one real parameter is left after imposing the anticommutation condition. This gives one zero mode doublet.

All together we obtain

$$(r-2)_{\text{long}} + 1_{\text{short}} + 1_{\text{Cartan}} = r$$

doublets of zero modes — the same number as for odd $r$.

We have proven the theorem:

**Theorem 4.** *In the theory with even $N$ with the fermion boundary conditions (3.7), the Dirac operator on the background $A_\mu(x, t)$ representing a pure gauge (2.10) [which corresponds to $B_\mu(x, t) = 0$] admits $\gcd(N, k)$ right and $\gcd(N, k)$ left zero modes in the topological sector $k$.*

As was mentioned above, the number of modes depends on the fermion boundary conditions. For example, if the periodic boundary conditions in the both directions are imposed, the number of the zero mode doublets is equal to [1]

$$n_0^{\text{double periodic}} = \gcd(N, k) - 1. \tag{3.22}$$

Indeed, we now have to count the elements of $\mathbb{c}$ that *commute* with $S$. This excludes the elements of the Cartan subalgebra. Consider a generic nondiagonal element of $\mathbb{c}$. It includes several ladders — long and short. The condition $[\Psi, S] = 0$ dictates that all the complex matrix elements in a long ladder coincide. The matrix elements in a short ladder also coincide with an additional constraint that they must be real. This gives the same count of parameters as in the problem with the boundary conditions (3.10). We arrive at (3.22).

If we choose the b.c. that are antiperiodic in imaginary time, but periodic in space, we need to count the matrices that commute with $S$ and anticommute with $C$. For $k = 1$, $S$ is related to $C$ by a group conjugation, $S = VCV^{-1}$ and the same is true for their centralizers. Thus, the centralizer of $S$ also represents a Cartan subalgebra embedded in $su(N)$ in a noncanonical way. Only one of its generators anticommutes with $C$, and we obtain one single zero mode doublet.

If $k > 1$, the matrices $S$ and $C$ belong to different conjugacy classes and there is no reason for the same counting in the problem where $S$ and $C$ are interchanged. And generically the counting is different, indeed. In the case $N = 4, k = 2$, it happens to be the same, but for $N = 6, k = 2$ it is already different: there are two matrices that satisfy $[\Psi, C] = \{\Psi, S\} = 0$ and no matrices satisfying $[\Psi, S] = \{\Psi, C\} = 0$ whatsoever [1].

We presented the calculation of the number of fermion zero modes on a torus in a particular gauge background. For $k = 1$, there is one doublet of zero modes or none depending on whether $N$ is even or odd. This conforms to the mod. 2 argument outlined above. But for higher $k$, the number of the zero mode doublets is sometimes larger than 1.

In order to understand whether the counting $n_0 = \gcd(N, k)$ holds for a generic field configurations, it is natural to choose some other handleable gauge background and solve the problem in that case.

# 4   Cartan instantons and their zero modes

In this section, we will not unwind the gauge field by the gauge transformation (2.6) and then deform it to zero, as we did before, but consider the original boundary conditions (2.4), choose the simplest topologically nontrivial gauge background $A_\mu(x, t)$ and study the Dirac spectrum there.

## 4.1   On the cylinder

A similar problem was first solved in Ref. [2]. In that paper, we studied the physics of the theory (1.1) at finite temperature, i.e. the theory was put on the cylinder with the finite extension $\beta$ along the imaginary time $t$ and the infinite spatial extension. We imposed the boundary conditions

$$\begin{aligned}
A_\mu(x, t + \beta) &= A_\mu(x, t), \\
\psi(x, t + \beta) &= -\psi(x, t).
\end{aligned} \tag{4.1}$$

and solved the Dirac equation in a topologically nontrivial background with $k = 1$. We obtained $N - 1$ doublets of zero modes. This result was then confirmed in [4], whose authors rotated the cylinder by $90^o$ and considered the theory on a finite spatial circle of length $L$ and the infinite extension in $t$. In [3], we generalised the discussion for any $k$ and discussed also the Dirac problem on an infinite Euclidean plane. We derived the presence of $k(N - k)$ doublets of zero modes.

We reproduce here this derivation following the approach of [4], where the physical instanton picture is somewhat more transparent. In the second half of this section, we translate it onto a finite torus.

Thus, we impose the boundary conditions[3]

$$
\begin{aligned}
A_\mu(x + L, t) &= A_\mu(x, t), \\
\psi(x + L, t) &= -\psi(x, t)
\end{aligned}
\tag{4.2}
$$

and impose the Hamilton gauge $A_0(x, t) = 0$. The instanton $A_1(x, t)$ is then interpreted as a topologically nontrivial tunneling transition trajectory between the different vacua, similar to the interpretation of the familiar BPST instanton [11].

We are in a position to study the vacuum structure of our theory. We consider the case $N = 2$ first.

In $QCD_2$, a classical vacuum with zero field density is the constant field configuration $A_1 = const$. By a gauge rotation, one can bring $A_1$ to the Cartan subalgebra. For $SU(2)$, we may pose $A_1 = a\tau^3$. Classically, the energy of all such constant configurations is zero. But taking into account the quantum corrections due to fermion loops,[4] the effective potential emerges [10]. In Refs.[10], bearing in mind the finite temperature applications, the effective potential for the *zeroth* component of the vector potential was calculated assuming the antiperiodic boundary conditions for the fermions under the Euclidean time shift. For $SU(2)$, this potential reads

$$
V^{\text{eff}}(A_0 = a\tau^3) = \frac{\beta}{2\pi} \left[ \left( 2a + \frac{\pi}{\beta} \right)_{\text{mod.} \frac{2\pi}{\beta}} - \frac{\pi}{\beta} \right]^2.
\tag{4.3}
$$

In our case, the dependence is the same:

$$
V^{\text{eff}}(a) = \frac{L}{2\pi} \left[ \left( 2a + \frac{\pi}{L} \right)_{\text{mod.} \frac{2\pi}{L}} - \frac{\pi}{L} \right]^2.
\tag{4.4}
$$

It is periodic with the period $\pi/(L)$ (see Fig. 1). This periodicity is due to the fact that any $a$ outside the interval

$$
0 \le a \le \frac{\pi}{L}
\tag{4.5}
$$

can be brought into this interval by a topologically *trivial* gauge transformation. There are two types of such transformations:

1. The shift $a \to a + (2\pi)/L$ realized by the gauge transformation

$$
\tilde{\Omega}(x) = \exp\left\{ \frac{2\pi i x}{L} \tau^3 \right\}.
\tag{4.6}
$$

   In contrast to the loop (4.7), this loop is contractible in $SU(2)/Z_2$.

2. The Weyl reflection $a\tau^3 \to -a\tau^3$ realized by the rotation by $\pi$ around the first or the second color axis.

---

[3] The physical picture happens to be more simple when the spatial fermionic boundary conditions are antiperiodic. It would also be interesting to perform a systematic study of the theory with periodic b.c. both for $A_\mu$ and $\psi$.

[4] In two dimensions, there are no physical degrees of freedom associated with the gauge fields, and the latter do not contribute.

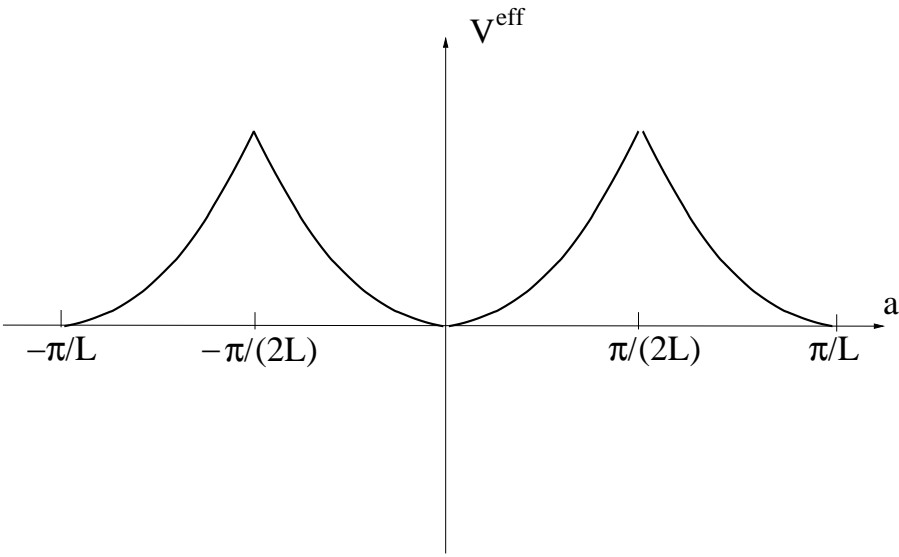

Figure 1: Effective potential (4.4)

The interval (4.5) may be called *Weyl alcove*.

We are left with only two vacuum states at $a = 0$ and $a = \pi/(L)$. They are related by a noncontractible gauge transformation

$$\Omega(x) = \exp\left\{\frac{i\pi x}{L}\tau^3\right\}, \qquad (4.7)$$

The instanton that we are interested in (we will call it the *Cartan* instanton) interpolates between these states along the path

$$A_1(t) = a(t)\tau^3 \quad \text{with} \quad a(-\infty) = 0, \ a(\infty) = \frac{\pi}{L}. \qquad (4.8)$$

We will assume that $a(t)$ tends to its asymptotic values exponentially fast.

Note that there are no antiinstantons: the configuration (4.8) with $a(-\infty) = \pi/L$ and $a(\infty) = 0$ belongs to the same topological class.

The Dirac equation (3.1) with $A_0 = 0$ and $A_1(t)$ given by (4.8) admits the following doublet of zero modes satisfying the antiperiodicity condition:

$$\Phi^+ = \tau^+ \begin{pmatrix} 0 \\ 1 \end{pmatrix} e^{i\pi x/L} e^{\phi(t)},$$

$$\Phi^- = \tau^- \begin{pmatrix} 1 \\ 0 \end{pmatrix} e^{-i\pi x/L} e^{\phi(t)}, \qquad (4.9)$$

where

$$\frac{d\phi(t)}{dt} = \frac{\pi}{L} - 2a(t). \qquad (4.10)$$

When $t \to \pm\infty$, the solutions behave as

$$\psi_\pm(t) \sim e^{-\pi|t|/L}. \qquad (4.11)$$

They are normalizable.

We see that there are two zero modes—with positive and negative chirality. They have the opposite color structure, which corresponds to the fields of positive and negative charge in the Abelian theory. In agreement with the standard Atiyah-Singer theorem, the zero modes have negative chirality in the former case and positive chirality in the latter case.

Consider now the case $N = 3$. The Weyl alcove for $SU(3)$ represents a triangle shown in Fig. 2. The effective potential for the classical vacuum,

$$A_1 = \mathrm{diag}(a_1, a_2, a_3), \qquad \sum_j a_j = 0,$$

is the sum of three terms:

$$V^{\mathrm{eff}}(a_j) = \frac{L}{2\pi} \sum_{j<k} \left[ \left( a_j - a_k + \frac{\pi}{L} \right)_{\mathrm{mod.}\frac{2\pi}{L}} - \frac{\pi}{L} \right]^2. \tag{4.12}$$

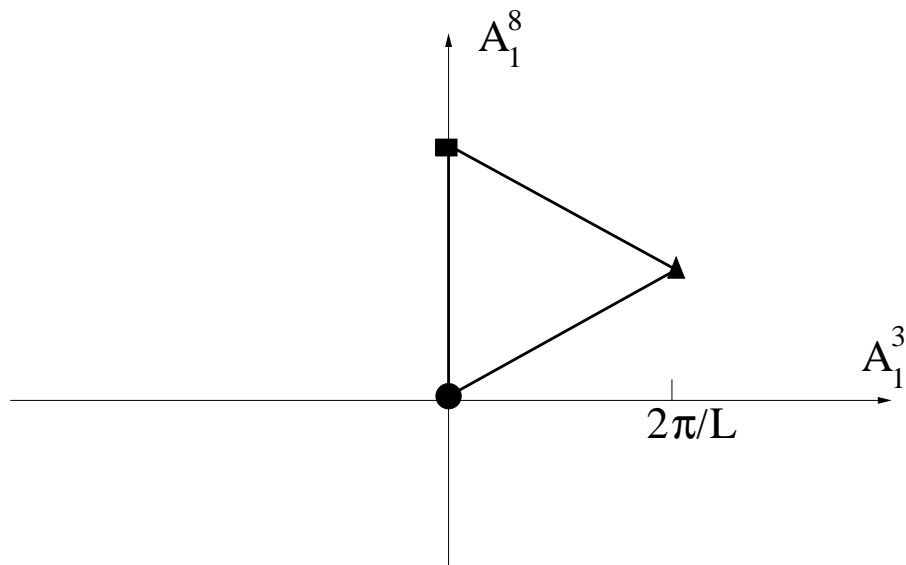

Figure 2: The Weyl alcove for $N = 3$. The vertices of the triangle mark the topologically distinct vacua.

The potential (4.12) has the minima at the vertices of the triangle:

$$A_1^{(0)} = 0; \quad A_1^{(1)} = \frac{2\pi}{3L}\mathrm{diag}(1, 1, -2) = \frac{4\pi}{L\sqrt{3}}t^8; \quad A_1^{(3)} = \frac{2\pi}{3L}\mathrm{diag}(2, -1, -1). \tag{4.13}$$

The Cartan instantons interpolate between different vacua. There are in principle 6 such configurations corresponding to three edges of the triangle passed in two directions, but all "clockwise" instantons belong to the same topological class $k = 1$ and all "counterclockwise"

instantons belong to the class $k = 2$. It is sufficient to consider only one of them, the configuration

$$A_1(t) = \frac{2}{3} a(t) \operatorname{diag}(1, 1, -2),$$ (4.14)

All other configurations have the same properties.

The Dirac equation (3.1) with $A_0 = 0$ and $A_1(t)$ given by (4.14) admits *two* doublets of zero modes:

$$\Phi_{1,2}^+ = E_{1,2}^+ \begin{pmatrix} 0 \\ 1 \end{pmatrix} e^{i\pi x/L} e^{\phi(t)},$$

$$\Phi_{1,2}^- = E_{1,2}^- \begin{pmatrix} 1 \\ 0 \end{pmatrix} e^{-i\pi x/L} e^{\phi(t)},$$ (4.15)

where

$$E_1^+ = \begin{pmatrix} 0 & 0 & 1 \\ 0 & 0 & 0 \\ 0 & 0 & 0 \end{pmatrix}, \qquad E_2^+ = \begin{pmatrix} 0 & 0 & 0 \\ 0 & 0 & 1 \\ 0 & 0 & 0 \end{pmatrix}$$ (4.16)

and $E_{1,2}^-$ are Hermitially conjugated. $\phi(t)$ is related to $a(t)$ in the same way (4.10) as before.

Generically, for larger values of $N$, the instantons of different types are labelled by an integer $k = 1, \ldots, N - 1$ and have the form

$$A_1(t) = \frac{2}{N} a(t) \operatorname{diag}(\underbrace{k, \ldots, k}_{N-k}, \underbrace{k - N, \ldots, k - N}_{k}).$$ (4.17)

Only the instantons with $k = 1, \ldots, [N/2]$ are essentially different. For example, for $SU(4)$, the Weyl alcove is an asymmetric tetrahedron with the vertices representing the following elements of $su(4)$: $O = 0$ and

$$A = \frac{\pi}{2L} \operatorname{diag}(1, 1, 1, -3); \quad B = \frac{\pi}{2L} \operatorname{diag}(2, 2, -2, -2); \quad C = \frac{\pi}{2L} \operatorname{diag}(3, -1, -1, -1).$$ (4.18)

It has four short and two long edges and, correspondingly, there are instantons of two types: with $k = 1$ and with $k = 2$.

The instanton (4.17) admits $k(N - k)$ zero mode doublets, given by the same formulas as in (4.15) with $k(N - k)$ different matrices $E^+$'s and $k(N - k)$ different $E^-$'s. Only one component of these matrices somewhere in the upper right (correspondingly, lower left) block $k \times (N - k)$ is equal to 1. All other components are zeros.

## 4.2 On the torus

We now roll up into a ring also the Euclidean time dimension and impose the boundary conditions (2.4) on the gauge fields and (3.7) on the fermion fields with a topologically nontrivial $\Omega(x)$.

Consider first the case $N = 2$. $\Omega(x)$ is given by (4.7). The instanton configuration reads

$$A_1(t) = \frac{\pi t}{L\beta} \tau^3.$$ (4.19)

Again, the Dirac operator admits a doublet of zero modes, which have, however, a somewhat more complicated form than on the cylinder,

$$
\begin{aligned}
\Phi^+ &= \tau^+ \begin{pmatrix} 0 \\ 1 \end{pmatrix} \sum_{n=-\infty}^{\infty} \exp\left\{ \frac{i\pi x}{L}(2n+1) \right\} \exp\left\{ -\frac{\pi\beta}{L}\left( \frac{t}{\beta} - n - \frac{1}{2} \right)^2 \right\}, \\
\Phi^- &= \tau^- \begin{pmatrix} 1 \\ 0 \end{pmatrix} \sum_{n=-\infty}^{\infty} \exp\left\{ -\frac{i\pi x}{L}(2n+1) \right\} \exp\left\{ -\frac{\pi\beta}{L}\left( \frac{t}{\beta} - n - \frac{1}{2} \right)^2 \right\}. \quad (4.20)
\end{aligned}
$$

These functions belong to the $\Theta$ family. The same functions describe the toric fermion zero modes in the constant magnetic field of unit flux in the Abelian theory.

It is also interesting to see what would happen if we imposed the periodic boundary conditions in both directions:

$$
\begin{aligned}
\psi(x+L,t) &= \psi(x,t), \\
\psi(x,t+\beta) &= \Omega(x)\psi(x,t)\Omega^{-1}(x), \quad (4.21)
\end{aligned}
$$

In this case, one obtains a doublet of the charged zero modes given by the similar expressions:

$$
\begin{aligned}
\Phi^+_{\text{per.}} &= \tau^+ \begin{pmatrix} 0 \\ 1 \end{pmatrix} \sum_{n=-\infty}^{\infty} \exp\left\{ \frac{2i\pi n x}{L} \right\} \exp\left\{ -\frac{\pi\beta}{L}\left( \frac{t}{\beta} - n \right)^2 \right\}, \\
\Phi^-_{\text{per.}} &= \tau^- \begin{pmatrix} 1 \\ 0 \end{pmatrix} \sum_{n=-\infty}^{\infty} \exp\left\{ -\frac{2i\pi n x}{L} \right\} \exp\left\{ -\frac{\pi\beta}{L}\left( \frac{t}{\beta} - n \right)^2 \right\}. \quad (4.22)
\end{aligned}
$$

And on top of that, there is a doublet of *constant neutral* zero modes:

$$
\Phi_1 = \tau^3 \begin{pmatrix} 1 \\ 0 \end{pmatrix} \quad \text{and} \quad \Phi_2 = \tau^3 \begin{pmatrix} 0 \\ 1 \end{pmatrix}. \quad (4.23)
$$

Two doublets altogether.

For higher $N$, the cylindrical argumentation above is translated onto the torus in a similar fashion. The toric zero modes have the same color structure as the cylindrical ones, while their coordinate dependence is the same as in (4.20).

We arrive at the conclusion:

*Naive instantons of the $SU(N)$ theory belonging to the topological sector $k$ admit $k(N-k)$ doublets of the fermion zero modes.*

## 5   Deformations

From the fact that different gauge background admit different number of zero modes, it follows that this number is not a topological invariant and may change under deformation. In this section, we confirm it in a quite explicit way. We will study the deformations of cylindrical Cartan instantons — this is essentially simpler than for the toric configurations.

## 5.1 Fixed asymptotics

To begin with, we will prove the following theorem:

**Theorem 5.** *In the gauge $A_0 = 0$, consider the gauge field background*

$$A_1(x,t) = A_1^{(0)}(t) + \alpha(x,t), \tag{5.1}$$

*where $A_1^{(0)}(t)$ is the Cartan instanton configuration (4.17) and $\alpha(x,t) = \alpha^a(x,t)t^a$ is periodic in $x$:*

$$\alpha(x,t) = \alpha^a(x,t)t^a = \sum_{m=-\infty}^{\infty} \alpha_m(t)e^{2\pi i m x/L} \tag{5.2}$$

*with $\alpha_n(t)$ falling off to zero as $t \to \pm\infty$ in such a way that the integral $\int \alpha_n(t)\,dt$ converges there.*

*Then the equation (3.1) still has $k(N-k)_L + k(N-k)_R$ normalized zero mode solutions in any order in the perturbation $\alpha(x,t)$.*

This statement and the idea of the proof can be found back in [2], but we did not give there much details, which we are going to provide now.

*Proof.* Consider first the $SU(2)$ theory.

Consider the fate of the positively charged mode [the first line in Eq.(4.9)]. This mode carries the negative chirality, and we can replace in this case $\sigma_1 \to 1$, $\sigma_2 \to -i$. We are going to solve the equation

$$\left(\frac{\partial}{\partial t} + i\frac{\partial}{\partial x}\right)\psi + [a(t)\tau^3 + \alpha, \psi] = 0 \tag{5.3}$$

by iterations. We pose

$$\psi = \psi_0 + \psi_1 + \psi_2 + \dots, \tag{5.4}$$

where $\psi_0$ is the Cartan zero mode in (4.9) and $\psi_n$ is of order $\sim \alpha^n$. We obtain the chain of equations

$$\left(\frac{\partial}{\partial t} + i\frac{\partial}{\partial x}\right)\psi_n + a(t)[\tau^3, \psi_n] = -[\alpha, \psi_{n-1}] \overset{\text{def}}{=} \gamma_{n-1}. \tag{5.5}$$

Each equation in (5.5) is split into three equations for the different color components:

$$\left(\frac{\partial}{\partial t} + i\frac{\partial}{\partial x}\right)\psi_n^3 = \gamma_{n-1}^3, \tag{5.6}$$

$$\left(\frac{\partial}{\partial t} + i\frac{\partial}{\partial x} + 2a(t)\right)\psi_n^+ = \gamma_{n-1}^+ \tag{5.7}$$

and

$$\left(\frac{\partial}{\partial t} + i\frac{\partial}{\partial x} - 2a(t)\right)\psi_n^- = \gamma_{n-1}^- \tag{5.8}$$

**I.** Let us assume first that the deformation $\alpha^a$ does not depend on $x$. Then the $x$-dependence of all the terms in (5.4) and (5.6)-(5.8) is the same as that of $\psi_0$

$$\psi_n(x,t) \;=\; \psi_n(t)e^{i\pi x/L}\,, \qquad \gamma_n^a(x,t) = \gamma_n(t)e^{i\pi x/L}\,. \tag{5.9}$$

In this case, the chain (5.6) - (5.8) is reduced to the system of ordinary differential equations,

$$\left(\frac{\partial}{\partial t} - \frac{\pi}{L}\right)\psi_n^3(t) \;=\; \gamma_{n-1}^3(t)\,, \tag{5.10}$$

$$\left(\frac{\partial}{\partial t} - \frac{\pi}{L} + 2a(t)\right)\psi_n^+(t) \;=\; \gamma_{n-1}^+(t)\,, \tag{5.11}$$

and

$$\left(\frac{\partial}{\partial t} - \frac{\pi}{L} - 2a(t)\right)\psi_n^-(t) \;=\; \gamma_{n-1}^-(t)\,. \tag{5.12}$$

We will prove the existence of the normalized solutions to this system for all $n$ by induction.

More exactly, we will prove that the correction $\psi_n(t)$ decays at $t \to \pm\infty$ as $\sim e^{-\pi|t|/L}$ at any order $n$.

- For $n = 0$, this follows from the explicit solution in (4.9).

- Suppose that $\psi_{n-1}(t)$ decays as $e^{-\pi|t|/L}$. Let us prove that $\psi_n(t)$ also has this property. Note first that, if $\psi_{n-1}(t) \sim e^{-\pi|t|/L}$, $\gamma_{n-1}(t)$ decays faster than $e^{-\pi|t|/L}$.

  Note also that, by our assumption about the behavior of $\alpha(t)$, the integrals $\int^{\infty} e^{\pi t/L}\gamma_{n-1}(t)\,dt$ and $\int_{-\infty} e^{-\pi t/L}\gamma_{n-1}(t)\,dt$ converge.

  *(i)* Consider Eq. (5.10). Its formal solution is

$$\psi_n^3(t) = -e^{\pi t/L}\int_t^{\infty} e^{-\pi t'/L}\,\gamma_{n-1}^3(t')\,dt' + C\,. \tag{5.13}$$

  Choose $C = 0$. For $t \to \infty$, $\psi_n^3(t)$ decays faster than $e^{-\pi t/L}$ together with $\gamma_{n-1}^3(t)$, and for $t \to -\infty$, the integral is finite and $\psi_n^3(t) \sim e^{\pi t/L} = e^{-\pi|t|/L}$.

  *(ii)*

  Consider now equation (5.11). Choose its particular solution in the form

$$\psi_n^+(t) \;=\; -e^{F(t)}\int_t^{\infty} e^{-F(t')}\,\gamma_{n-1}^+(t')\,dt'\,, \tag{5.14}$$

  where

$$F'(t) = \frac{\pi}{L} - 2a(t)\,. \tag{5.15}$$

  When $t \to \infty$, $F(t) \to -\pi t/L$ and

$$\psi_n^+(t) \;\sim\; e^{-\pi t/L}\int_t^{\infty} e^{\pi t'/L}\,\gamma_{n-1}^+(t')\,dt' \;\sim\; e^{-\pi t/L}\,,$$

as the integral converges at the upper limit.

When $t \to -\infty$, $F(t) \to \pi t/L = -\pi|t|/L$ and

$$\psi_n^+(t) ~\sim~ e^{-\pi|t|/L} \int_{-\infty}^{\infty} e^{\pi t'/L} \gamma_{n-1}^+(t')\, dt' ~\sim~ e^{-\pi|t|/L}\,,$$

as the integral converges at both limits.

*(iii)*

For Eq. (5.12), the reasoning is analogous. We choose its solution as

$$\psi_n^-(t) ~=~ -e^{G(t)} \int_t^{\infty} e^{-G(t')} \gamma_{n-1}^-(t')\, dt'\,, \tag{5.16}$$

where $G'(t) = \frac{\pi}{L}[1 + 2a(t)]$.

When $t \to \infty$,

$$\psi_n^-(t) ~\sim~ e^{3\pi t/L} \int_t^{\infty} e^{-3\pi t'/L} \gamma_{n-1}^-(t')\, dt' ~\sim~ e^{-\pi t/L}\,.$$

When $t \to -\infty$,

$$\psi_n^-(t) ~\sim~ e^{-\pi|t|/L} \int_{-\infty}^{\infty} e^{-G(t')} \gamma_{n-1}^-(t')\, dt' ~\sim~ e^{-\pi|t|/L}\,.$$

**II**. Let us now take into account the higher Fourier modes in the expansion (5.2).

Their presence entails the presence of higher Fourier modes in the expansion for the correction $\psi_n(x,t)$:

$$\psi_n(x,t) ~=~ \sum_m \psi_n^{(m)}(t) \exp\left\{ \frac{i\pi x}{L}(1 + 2m) \right\} \tag{5.17}$$

The equations (5.10) - (5.12) acquire the form

$$\left( \frac{\partial}{\partial t} - \frac{\pi}{L}(2m + 1) \right) \psi_n^{(m)\,3}(t) ~=~ \gamma_{n-1}^{(m)\,3}(t)\,, \tag{5.18}$$

$$\left( \frac{\partial}{\partial t} - \frac{\pi}{L}(2m + 1) + 2a(t) \right) \psi_n^{(m)\,+}(t) ~=~ \gamma_{n-1}^{(m)\,+}(t)\,, \tag{5.19}$$

and

$$\left( \frac{\partial}{\partial t} - \frac{\pi}{L}(2m + 1) - 2a(t) \right) \psi_n^{(m)\,-}(t) ~=~ \gamma_{n-1}^{(m)\,-}(t)\,, \tag{5.20}$$

where

$$\gamma_{n-1}^{(m)\,a}(t) ~=~ -\sum_{p+q=m} [\alpha_p(t), \psi_{n-1}^{(q)}(t)]\,. \tag{5.21}$$

We can prove now that the three components of $\psi_n^{(m)}(t)$ exponentially decay at large $|t|$ by induction in the same way as we did it in the absence of the higher harmonics in (5.2). Consider e.g. Eq. (5.18). Let $m > 0$. Choose the particular solution of the equation in the form

$$\psi_n^{(m)\,3}(t) = -e^{\pi t(2m+1)/L} \int_t^{\infty} e^{-\pi t'(2m+1)/L} \gamma_{n-1}^{(m)\,3}(t')\, dt'\,. \tag{5.22}$$

By inductive assumption, $\psi_{n-1}^{(m)\,a}(t)$ and hence $\gamma_{n-1}^{(m)\,a}(t)$ fall down $\sim e^{-\pi|t|/L}$. By the same reasoning as before, it follows that $\psi_n^{(m)\,3}(t)$ falls down $\sim e^{-\pi|t|/L}$; the presence of the factor $2m+1$ in the exponents in Eq.(5.22) is irrelevant.

If $m < 0$, we choose the solution in the form

$$\psi_n^{(m)\,3}(t) = e^{\pi t(2m+1)/L} \int_{-\infty}^{t} e^{-\pi t'(2m+1)/L} \gamma_{n-1}^{(m)\,3}(t')\, dt' \tag{5.23}$$

and, by exploring the limits $t \to \pm\infty$, deduce that it falls down $\sim e^{-\pi|t|/L}$.

The equations (5.19) and (5.20) can be treated in a similar way.

### 5.1.1   $N > 2$

This proof can be translated without much change to the theories with higher $N$. Consider the $SU(3)$ theory. The Cartan instanton has the form (4.14). It has two doublets of zero modes. Let us add the deformation (5.2) with the same properties as before and explore the fate of one of the positive chirality modes. For example, the fate of the mode

$$\Phi_1^{+(0)}(t) \;=\; \begin{pmatrix} 0 & 0 & 1 \\ 0 & 0 & 0 \\ 0 & 0 & 0 \end{pmatrix}_{\text{color}} \begin{pmatrix} 0 \\ 1 \end{pmatrix}_{\text{spin}} \Phi^{(4+i5)(0)}(t)\,,$$

$$\tag{5.24}$$

with

$$\Phi^{(4+i5)(0)}(t) \;=\; e^{i\pi x/L} e^{\phi(t)}, \qquad \phi'(t) = \pi[1 - 2a(t)]/L\,.$$

We obtain the chain of the equations

$$\left( \frac{\partial}{\partial t} - \frac{\pi}{L}(2m+1) \right) \psi_n^{(m)\,1,2,3,8}(t) \;=\; \gamma_{n-1}^{(m)\,1,2,3,8}(t) \tag{5.25}$$

and

$$\left( \frac{\partial}{\partial t} - \frac{\pi}{L}(2m+1) \mp \frac{2\pi}{L}a(t) \right) \psi_n^{(m)\,4\pm i5}(t) \;=\; \gamma_{n-1}^{(m)\,4\pm i5}(t)\,,$$

$$\left( \frac{\partial}{\partial t} - \frac{\pi}{L}(2m+1) \mp \frac{2\pi}{L}a(t) \right) \psi_n^{(m)\,6\pm i7}(t) \;=\; \gamma_{n-1}^{(m)\,6\pm i7}(t) \tag{5.26}$$

with

$$\gamma_{n-1}^{(m)\,a}(t) \;=\; if^{abc} \sum_{p+q=m} \alpha_p^c(t)\psi_{n-1}^{(q)\,b}(t)\,. \tag{5.27}$$

The inductive proof that, at any order, the corrections $\psi_n^{(m)\,a}(t)$ fall down exponentially as $t \to \pm\infty$ is translated from the proof for the $N = 2$ theory without much change.

For an arbitrary $N$, we obtain a similar chain. Ii involves the equations for the components $\psi_n^{(m)\,a}(t)$ where the index $a$ corresponds to the centralizer $SU(k) \times SU(N-k) \times U(1)$ of the Cartan instanton confuguration (4.17). These components do not "feel" the presence of the gauge field. It involves also $k(N-k)$ doublets of the components corresponding to the root vectors that do not commute with (4.17). The inductive proof constructed above works also in this case.

$\square$

## 5.2 Generic deformations

To understand that the deformations considered above are not the most generic ones, consider the $SU(2)$ theory with the gauge background

$$A_1(t) = \tau^3 b(t), \qquad \text{with} \qquad b(-\infty) = 0, \quad b(\infty) = \frac{2\pi}{L}. \tag{5.28}$$

In this case, the Dirac equation admits *two* doublets of normalized zero modes. The positively charged modes are

$$\Phi_1(x,t) = \tau^+ \begin{pmatrix} 0 \\ 1 \end{pmatrix} e^{i\pi x/L} e^{\phi_1(t)},$$

$$\Phi_2(x,t) = \tau^+ \begin{pmatrix} 0 \\ 1 \end{pmatrix} e^{3i\pi x/L} e^{\phi_2(t)}, \tag{5.29}$$

where

$$\frac{d\phi_1}{dt} = \frac{\pi}{L} - 2b(t), \qquad \frac{d\phi_2}{dt} = \frac{3\pi}{L} - 2b(t). \tag{5.30}$$

In the Abelian theory, the existence of two zero modes follows from the Atiyah-Singer theorem: the field (5.28) carries the double magnetic flux. However, in the non-Abelian theory, the configuration (5.28) is topologically trivial, it satisfies the boundary condition

$$A_1(x,\infty) = -i\partial_x \tilde{\Omega}(x)\,\tilde{\Omega}^{-1}(x) + \tilde{\Omega}(x) A_1(x,-\infty)\tilde{\Omega}^{-1}(x) \tag{5.31}$$

with a contractible loop (4.6). We expect that the zero modes (5.29) are not robust under a generic non-Abelian deformation.

However, they *are* robust under the deformations of the same kind as in (5.1) with the requirement that $\alpha(x,t)$ vanishes rapidly at $t = \pm\infty$. All the steps in the proof of Theorem 5 can also be reproduced in this case. The deformations of a more general nature such that the corresponding Dirac operator does not sustain zero modes anymore must exist, and they *do*.

The loop (4.6) is contractible, i.e. there exists a continuous family $\Omega_\xi(x)$ such that

$$\Omega_\xi(0) = \Omega_\xi(L) = \mathbb{1}, \quad \Omega_0(x) = \Omega(x), \qquad \Omega_1(x) = \mathbb{1}. \tag{5.32}$$

Consider the corresponding family of field configurations,

$$A_1^{(\xi)}(x,t) = -i\frac{b(t)L}{2\pi}\partial_x \Omega_\xi(x)\,\Omega_\xi^{-1}(x). \tag{5.33}$$

As $\xi$ changes from 0 to 1, the configuration (5.33) interpolates between the field (5.28) sustaining two doublets of zero modes to the configuration $A_1(x,t) = 0$ where the zero modes are absent[5] A very plausible guess is that the zero modes disappear as soon as $\xi \neq 0$ and the loop slides aside as in Fig. 3.

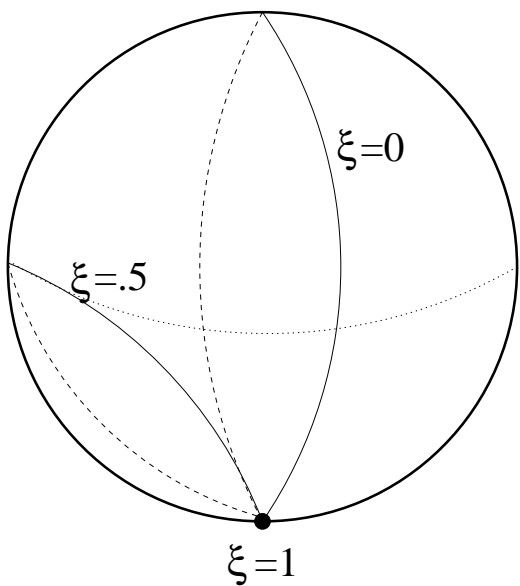

Figure 3: Sliding loops

Note that the configuration

$$A_1^{(\xi)}(x,\infty) \;=\; -i\partial_x\Omega_\xi(x)\,\Omega_\xi^{-1}(x) \tag{5.34}$$

is still a vacuum configuration with zero field density [we assumed that $b(t \to \infty)$ approaches the value $2\pi/L$ exponentially fast]. It is topologically trivial, being related to $A_1 = 0$ and to $A_1 = 2\pi\tau^3$ by contractible gauge transformations. But the configurations (5.33) are *not* related to (5.28) by gauge transformations. They represent genuine deformations of (5.28), and these deformations matter!

Note also that one *can* find a gauge transformation $A_1 \to \tilde{A}_1$ that brings (5.33) to the form where $\tilde{A}_1(x,t)$ keeps its asymptotic values: $\tilde{A}_1^{(\xi)}(x,-\infty) = 0$ and $\tilde{A}_1^{(\xi)}(x,\infty) = 2\pi\tau^3$. This transformation is realized by $\omega_\xi(x,t)$ such that

$$\omega_\xi(x,-\infty) = \mathbb{1}, \qquad \omega_\xi(x,\infty) = \tilde{\Omega}(x)\Omega_\xi^{-1}(x)\,.$$

Thus, the fact that the configuration (5.33) carries no zero modes if $\xi \neq 0$ seems to contradict the statement above that the zero modes are robust under the deformations that keep the asymptotics. There is no contradiction, however, as the transformation $\omega_\xi(x,t)$ brings about a nonzero $A_0(x,t)$. And if one imposes the Hamilton gauge, a generic deformed instanton configuration does not keep the simple boundary condition assumed in Sect. 5.1.

---

[5]A constant zero mode for the free Dirac equation would be present if $\psi(x,t)$ satisfied *periodic* boundary conditions on the spatial circle, but our b.c. are *anti*periodic.

For $N = 2$, a generic topologicallly nontrivial instanton keeps a single doublet of zero modes. This is dictated by Theorem 2, which is also valid on a cylinder. But the two doublets of zero modes (4.15) for $N = 3$ theory are not robust under a generic deformation. To understand that, consider along with the field (4.14),

$$A_1(t) = \frac{2}{3}a(t)\operatorname{diag}(1, 1, -2),\tag{5.35}$$

the field

$$B_1(t) = \frac{4}{3}a(t)\operatorname{diag}(2, -1, -1)\tag{5.36}$$

These two fields satisfy the same boundary conditions and belong to the same topological class. The vacua at $t = \infty$ are related as

$$B_1(\infty) = -i\partial_x V(x)V^{-1}(x) + V(x)A_1(\infty)V^{-1}(x),\tag{5.37}$$

where

$$V(x) = \exp\left\{\frac{2i\pi x}{L}\operatorname{diag}(1, -1, 0)\right\}\tag{5.38}$$

represents a contractible loop in $SU(3)/Z_3$.

Still, the configuration (5.35) has two doublets of zero modes and the configuration (5.36) has *four* such doublets: the modes

$$\begin{pmatrix} 0 & 1 & 0 \\ 0 & 0 & 0 \\ 0 & 0 & 0 \end{pmatrix}_{\text{color}} \begin{pmatrix} 0 \\ 1 \end{pmatrix}_{\text{spin}} e^{i\pi x/L}e^{\phi_1(t)}, \qquad \begin{pmatrix} 0 & 0 & 1 \\ 0 & 0 & 0 \\ 0 & 0 & 0 \end{pmatrix}_{\text{color}} \begin{pmatrix} 0 \\ 1 \end{pmatrix}_{\text{spin}} e^{i\pi x/L}e^{\phi_1(t)}\tag{5.39}$$

with $\dot{\phi}_1(t) = \pi/L - 4a(t)$,

$$\begin{pmatrix} 0 & 1 & 0 \\ 0 & 0 & 0 \\ 0 & 0 & 0 \end{pmatrix}_{\text{color}} \begin{pmatrix} 0 \\ 1 \end{pmatrix}_{\text{spin}} e^{3i\pi x/L}e^{\phi_2(t)}, \qquad \begin{pmatrix} 0 & 0 & 1 \\ 0 & 0 & 0 \\ 0 & 0 & 0 \end{pmatrix}_{\text{color}} \begin{pmatrix} 0 \\ 1 \end{pmatrix}_{\text{spin}} e^{3i\pi x/L}e^{\phi_2(t)}\tag{5.40}$$

with $\dot{\phi}_2(t) = 3\pi/L - 4a(t)$, and the Hermitially conjugated modes of opposite chirality.

Consider now the family of configurations $A_1^{(\xi)}(x, t)$ that smoothly interpolate between $A_1(t)$ and $B_1(t)$ as $\xi$ changes from 0 to 1. For intermediate values of $\xi$, there probably are no zero modes at all.

# 6 Discussion: screening vs. confinement

The technical issue about the existence or non-existence of fermion zero modes in adjoint $QCD_2$ is interesting by its relationship to the physics of this model. The model is confining in the sense that its physical spectrum does not involve states that carry color charge. But a nontrivial question is whether we are dealing with confinement in the strong sense where the potential between two static color sources grows linearly and the Wilson loop has the area law or with the confinement in the weak sense or *screening* with the perimeter law for the Wilson loop.

For example, in the pure Yang-Mills theory in 4 dimensions, we have[6] strong confinement of fundamental heavy sources, but adjoint sources are screened by the gluons. In $QCD_4$, the fundamental sources are also screened due to the presence of dynamical fundamental quarks. The ordinary $QCD_2$ with dynamical quarks has the same properties as $QCD_4$: any heavy colored source is screened. The adjoint $QCD_2$ with massive fermions has the same properties as 4-dimensional gluodynamics: strong confinement for fundamental sources, whereas adjoint sources (and all other sources with zero $n$-ality) are screened.

And the *massless* adjoint $QCD_2$ exhibits a nontrivial behavior. We showed in Ref. [13] that, contrary to naive expectations, the massless adjoint fermions may well screen the fundamental sources and all other sources with nonzero $n$-ality.[7] This observation was based on the following conjecture: [8]

**Conjecture 1.** *In the topological trivial sector of the massless adjoint $QCD_2$, the fundamental Wilson loop average*

$$\langle W(C) \rangle_{\mathrm{triv}} \;=\; \left\langle \frac{1}{N} \mathrm{Tr} \left\{ P \exp \left( ig \oint_C A_\mu(x)\, dx^\mu \right) \right\} \right\rangle_{\mathrm{triv}} \tag{6.1}$$

*satisfies the property*

$$\sigma \;=\; \lim_{\mathcal{A} \to \infty} \left[ -\frac{\ln \langle W(C) \rangle_{\mathrm{triv}}}{\mathcal{A}} \right] \;=\; 0\,, \tag{6.2}$$

*where $\mathcal{A}$ is the area embraced by the contour.*

In other words, the averages of large Wilson loops fall down in this sector according to the perimeter law rather than area law.

Together with most other experts, we believe this conjecture to be correct though, to the best of our knowledge, its formal proof has not been given yet. The difficulty lies in the non-Abelian nature of the theory. The Abelian version of this conjecture is an exact theorem:

*Proof.* Consider the Schwinger model at the infinite Euclidean plane in the trivial topological sector,

$$\Phi = \frac{e}{2\pi} \int F(x)\, d^2 x \;=\; 0\,, \tag{6.3}$$

where $F = \varepsilon_{\mu\nu} \partial_\mu A_\nu / 2$ is the Abelian field density and the fermion charge $e$ is included in the definition of the flux. Suppose that $F(x)$ vanishes at infinity fast enough. In view of (6.3), one can then choose a gauge where $A_\mu(x)$ also vanishes there. And that means that the Abelian Wilson loop,

$$W^{\mathrm{Ab}}(C) \;=\; \exp \left( ie' \oint_C A_\mu(x)\, dx^\mu \right)\,, \tag{6.4}$$

---

[6]Or rather we *believe* to have.

[7]A similar phenomenon is known to take place in the massless Schwinger model where the massless fermions of charge one manage to screen any heavy source of integer or fractional electric charge [12].

[8]In Ref. [13], we also presented several other arguments, but we only discuss here the most solid one having an immediate relationship to the main subject of this paper.

is equal to 1 for the asymptotic contour $C$ embracing infinity, and this is true for any charge $e'$ of a heavy probe.

Note that in the sectors with nonzero flux $\Phi = n$, the vector potential behaves at infinity as

$$A_\mu(x) = -i[\partial_\mu e^{in\theta}] e^{-in\theta}, \tag{6.5}$$

where $\theta$ is the polar angle, and the asymptotic Wilson loop takes the value

$$W_n^{\text{asympt}} = \exp\left\{\frac{2i\pi ne'}{e}\right\}.$$

We go back to the sector $n = 0$ and consider a large but not asymptotically large loop. The Stokes theorem allows us to write

$$\langle W^{\text{Ab}}(C)\rangle_{n=0} = \left\langle \exp\left\{ie' \int_D F(x)\, d^2x\right\}\right\rangle_{n=0}, \tag{6.6}$$

where $D$ is the domain embraced by the loop. The Gaussian nature of the path integral in the Schwinger model allows one to present it as

$$\exp\left\{-\frac{e'^2}{2} \int_D \int_D d^2x\, d^2y\, \langle F(x)F(y)\rangle_{n=0}\right\}. \tag{6.7}$$

The correlator $\langle F(x)F(y)\rangle_{n=0}$ depends only on $x - y$. In the massless Schwinger model, it is known exactly. It decays exponentially $\propto e^{-\mu\sqrt{(x-y)^2}}$ at large separations ($\mu = e/\sqrt{\pi}$ being the mass of the Schwinger boson) and satisfies the property

$$\int_{\text{whole plane}} d^2x\, \langle F(x)F(0)\rangle_{n=0} = 0. \tag{6.8}$$

If the integral (6.8) did not vanish, the exponent in (6.7) would be proportional to the area $\mathcal{A}$ of the domain $D$ giving a nonzero string tension. But as it vanishes, the string tension vanishes too.

If the integral is done over a finite region $D$ rather than the whole plane, the integrals in (6.8), (6.7) do not vanish. The double integral in (6.7) is saturated by the values of $x$ that are close to the border of $D$ and $y$ that are within the distance $\sim \mu^{-1}$ from $x$. We obtain the perimeter law [14],

$$\langle W^{\text{Ab}}(C)\rangle_{n=0} = \exp\left\{-\frac{e'^2 P}{4\mu}\right\}. \tag{6.9}$$

$\square$

Consider now the non-Abelian theory of interest. Assuming as before that the field density vanishes at infinity, the gauge potential in the topological sector $k$ can be brought into the form

$$A_\mu(x) = -i\left[\partial_\mu \exp\left\{\frac{ik\theta}{N}\tau^3\right\}\right] \exp\left\{-\frac{ik\theta}{N}\tau^3\right\}, \tag{6.10}$$

This gives the values

$$W_k = e^{2i\pi k/N} \tag{6.11}$$

for the asymptotic Wilson loop [6]. For $k = 0$, it is just the unity.

The property (6.2) could possibly be proven using the non-Abelian version of the Stokes theorem. The latter reads [15]

$$P \exp \left\{ i \oint_C A_\mu(x) \, dx^\mu \right\} = \mathcal{P} \exp \left\{ i \int_D \mathcal{F} \, d^2 x \right\}, \tag{6.12}$$

where $P$ is the ordinary path ordering and $\mathcal{P}$ is the operator of *area-ordering*, i.e. the infinitesimal loops should first be multiplied over along the direction of $x$, and then along the direction of $y$. As for $\mathcal{F}$, it is not simply the field density, but the object

$$\mathcal{F}(x) = U F(x) U^{-1}, \tag{6.13}$$

where

$$U = P \exp \left\{ i \int_O^x A_\mu(x) \, dx^\mu \right\} \tag{6.14}$$

is the string operator connecting a reference point $O$ on the contour $C$ to the point $x$ inside the contour along a certain prescribed path.

Anyway, if we assume that the conjecture above is correct, we can be sure that the $N = 2$ theory exhibits screening. Indeed, the topologically nontrivial sector there *has* a doublet of zero modes, the fermion determinant vanishes and hence this sector does not contribute to the path integral. Whatever is true in the topologically trivial sector (like the perimeter law for the Wilson loop), is true in the whole theory.

The situation is less clear for higher $N$ starting from $N = 3$. In [13], the existence of the zero modes for $k \neq 0$ was assumed also for higher $N$, which entailed the conclusion that this theory exhibits screening. But we know today that, e.g. for $N = 3$, a generic Euclidean field configuration does not sustain fermion zero modes and the contributions of the topologically nontrivial sectors do not vanish. In the sum of the contributions of the sectors $k = 0, 1, 2$, cancellations might occur, so that $\langle W(C) \rangle$ exhibits the area rather than perimeter law and the theory is confining [1].

In recent [16], it was argued, however, that this does not happen and adjoint massless $QCD_2$ with any unitary gauge group always exhibits screening. String operators similar to (6.14) played an important role in this analysis.

An independent argument in favour of the screening scenario was given in [17].

More studies in this direction are highly desirable.

# Acknowledgements

I am indebted to Igor Klebanov, Zohar Komargodsky and Yuya Tanizaki for illuminating discussions.

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
