# Peer review of "A comment on instantons and their fermion zero modes in adjoint $QCD_2$."

_SciPost Physics_

## Round 1 · Referee Report · Anonymous (Referee 1) · 2021-6-1

Strengths

The strength of this paper lies in the explicit nature of its arguments, somewhat of a rarity in the modern discourse on this and related subjects. The zero modes of the Dirac equation in 't Hooft flux (my name, not the author's) backgrounds are studied in detail and intuition for the mod(2) index theorem is provided in a clear and explicit manner. I think this makes the paper is of value, especially to a less mathematically oriented physics audience.

The implications of the results re. confinement in adjoint YM theory in two dimensions are discussed and the current state of the arguments on the subject are also clearly presented.

Weaknesses

I do not see any weaknesses.

Report

As far as I can tell, yes.

Requested changes

I am not requesting any changes, the paper clearly states its goals and results as well as its implications.

---

## Round 1 · Referee Report · Anonymous (Referee 2) · 2021-6-8

Strengths

Very explicit calculations.

Weaknesses

The final conclusion section could do a better job of summarising the literature.

Report

I like this paper. It studies the centre of modes of Z_N center vortices in 2d adjoint QCD. These are protected by a mod 2 index theorem, rather than the more familiar Atiyah-Singer index theorem .

In many ways, the calculation is rather old-fashioned, explicitly solving the Dirac equation in various background field configurations and showing how zero modes can be lifted as the field is deformed, always preserving the mod 2 index. However the old-fashioned methodology of the paper is also its strength: no fancy topological arguments are needed and all the results are presented in a very clear and straightforward way.

My only small complaint is that I was somewhat disappointed by the conclusions section. A rather complicated story has arisen in the past two years. The papers from the 90s (the most important of which was co-authored by the current author) suggested, rather strikingly, that fundamental Wilson lines are screened in massless adjoint QCD2. More recently, in 2019, Cherman et al argued that this conclusion is not correct, a claim that was motivated in part by the mod 2 index of fermi zero modes that are the subject of the present paper. Subsequently, work by Komargodski et al and by Dempsey et al argued that the previous results were correct after all, albeit with new insights by incorporating some important results from the Cherman et al paper.

The upshot is that the current situation is a little baffling for a casual reader. In particular, the present paper confirms the mod 2 calculation of the Cherman et al paper, but disagrees with their conclusions (assuming that I myself am not confused which might be a big assumption!). The author doesn't take the reader through the clearest path through this morass of conflicting claims, and it's not clear to me how the mod 2 index calculation (which is clearly correct) can be wielded, judo fashion, to argue against the very authors who first proposed it. It would be nice to get a clearer picture of this in the conclusion section.

Requested changes

I think the paper could be improved by a clearer conclusion section.

---

## Editorial Decision

resubmitted